Review

# FERM domain–containing proteins are active components of the cell nucleus

Péter Borkúti[1], Ildikó Kristó[1], Anikó Szabó[1], Zoltán Kovács[1,2], Péter Vilmos[1]

**The FERM domain is a conserved and widespread protein module that appeared in the common ancestor of amoebae, fungi, and animals, and is therefore now found in a wide variety of species. The primary function of the FERM domain is localizing to the plasma membrane through binding lipids and proteins of the membrane; thus, for a long time, FERM domain–containing proteins (FDCPs) were considered exclusively cytoskeletal. Although their role in the cytoplasm has been extensively studied, the recent discovery of the presence and importance of cytoskeletal proteins in the nucleus suggests that FDCPs might also play an important role in nuclear function. In this review, we collected data on their nuclear localization, transport, and possible functions, which are still scattered throughout the literature, with special regard to the role of the FERM domain in these processes. With this, we would like to draw attention to the exciting, new dimension of the role of FDCPs, their nuclear activity, which could be an interesting novel direction for future research.**

## The FERM Domain and Its History

The FERM domain (F for protein 4.1, E for ezrin, R for radixin, and M for moesin) is about 30 kD and is found in a number of cytoskeletal proteins that bind plasma membrane proteins. The crystal structure of the FERM domain reveals that it has a tripartite organization that forms a compact clover-shaped structure (Hamada et al, 2000) (Fig 1A). The N-terminal F1 module consists of a ubiquitin-like fold, the F2 lobe forms an acyl-CoA–binding protein fold, and the C-terminal F3 module is structurally similar to the pleckstrin homology (PH) domain that is known to bind phosphorylated phosphoinositides. The FERM domain as a cysteine-rich, basic-charged globular module enables protein–protein and protein–lipid interactions (Fig 1B), which determine the activity and specificity of the given FERM domain–containing protein (FDCP). The most important binding molecules of the FERM domain are transmembrane receptors, integrins, IP3, and PtdIns(4,5)P2 (PIP2).

In the human genome, more than 30 genes encode proteins containing FERM domains. A comparison of FERM domains of 185 FDCPs identified four major groups (Ali & Khan, 2014) (Fig 2). The four broad groups represent proteins with very different functions relative to each other, and often even the members within the groups. According to genome sequence data, the FERM domain first appeared in the ancestors of today's myosin and talin proteins during the separation of plants and Amorphea (amoebae, fungi, and animals), about 1.4 billion years ago (Ali & Khan, 2014) (Fig 3). Today, the functions of both myosins and talins are closely related to actin, so it can be assumed that the FERM domain contributed to increasing the complexity of the actin cytoskeleton; its original function may have been to anchor proteins or lipids to the microfilaments. The FERM domain probably played an important role also in the evolution of multicellularity, because along with the appearance of multicellular animals (metazoa), ERM and P4.1 proteins also appeared as new actin-organizing players, and the FERM domain began to be used for new functions. Although protein kinases, kindlin and E3-MYLIP, no longer anchor other proteins to actin, they are involved in the organization of the cytoskeleton and cell–cell junction structures. Along with the development of bilaterians, divergence in the application of the FERM domain has reached a new level. In addition to Merlin, a new member of FDCPs that cross-link actin (Michie et al, 2019), new kinases (JAK), protein phosphatases (PTNs), and FRPD and FRMD proteins also emerged (Fig 3). The activities of the newest FDCPs are no longer primarily related to the cytoskeleton, but because some of their functions are still associated with it, we can say that the FERM domain has become a versatile tool for increasingly specific and diverse cytoskeletal functions.

Today, we know that all the most important components of the cytoskeleton are also present in the nuclear compartment (Kumeta et al, 2012; Kristó et al, 2016; Bajusz et al, 2018); therefore, FDCPs cooperating with them are also likely to be found in the nucleus. Most of the literature data on nuclear localization refer to Merlin and FAK proteins, but with the exception of MyoVII, MyoXV, and E3-MYLIP, also to the surprise of the authors of the present study, despite their apparently primarily cytoskeletal activity, almost all members of the large group of FDCPs are found in the cell nucleus (Table 1).

[1]HUN-REN Biological Research Centre, Szeged, Hungary   [2]Doctoral School of Multidisciplinary Medical Science, University of Szeged, Szeged, Hungary

Correspondence: vilmosp@brc.hu

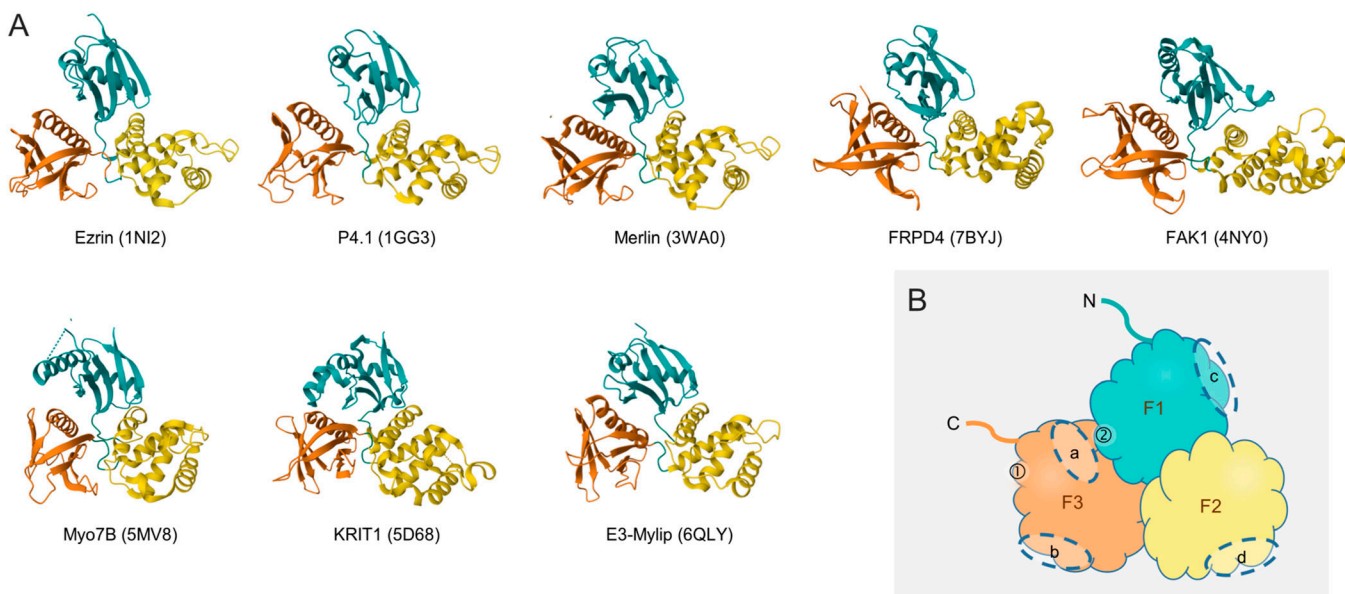

**Figure 1. Conserved structure of the FERM domain.**
**(A)** Crystal structures of the FERM domains of human FERM domain–containing proteins representing all FERM domain–containing protein groups demonstrate the high evolutionary conservation of the FERM domain. PDB accession numbers are in parentheses: 1NI2 (Smith et al, 2003), 1GG3 (Han et al, 2000), 3WA0 (Mori et al, 2014), 7BYJ (Wang et al, 2020), 4NY0 (Brami-Cherrier et al, 2014), 5MV8 (Yu et al, 2017), 5D68 (Zhang et al, 2015), and 6QLY (Martinelli et al, 2020). **(B)** Cartoon depiction of the general structure of the FERM domain. The main binding sites identified in ERM proteins are labeled. ①—transient IP3/PIP2-binding site; ②—stable IP3/PIP2-binding site; and (a–d)—protein–protein interaction sites. Binding partners are for example in the case of (a)—moesin + Crumbs (PDB accession number 4YL8 [Wei et al, 2015]); (b)—radixin + EBP50 (2D10 [Terawaki et al, 2006]); (c)—radixin + MT1-MMP (3X23 [Terawaki et al, 2015]); and (d)—Merlin + Lats1 (4ZRK [Li et al, 2015]).

## ERM proteins

ERM proteins, after which the FERM domain was named, anchor membrane proteins to the actin network (Jankovics et al, 2002). Initial observation suggesting nuclear and even nucleolar localization for ezrin emerged early (Kaul et al, 1999). Later, genome-wide proteome analysis (Bergquist et al, 2001) and immunocytochemical studies (Melendez-Vasquez et al, 2001) detected human ERMs and the single Drosophila ERM (Batchelor et al, 2004) in the nucleus. Their amount in the nucleus is regulated by cell density, hormonal treatment, and cellular stress such as heat shock, which suggests controlled nuclear transport (Batchelor et al, 2004; Kristó et al, 2017). Accordingly, an NLS motif outside the FERM domain ($R_{435}RRK$ in human ezrin) has been identified in mammalian ERMs (Batchelor et al, 2004); however, this motif is conserved in vertebrates only. Another canonical NLS motif (RRRK/R) has been in silico–predicted at the C-terminal end of the FERM domain (Krawetz & Kelly 2008), which shows a very high degree of evolutionary conservation (Fig 4A), so it is conceivable that it is a functional NLS. The region around this potential NLS has also been described in ezrin as a monomeric actin-binding site (Roy et al, 1997), suggesting the possibility that similar to the actin-binding transcription cofactors, MRTF-A (Miranda et al, 2021) and JMY (Zuchero et al, 2012), the intracellular polymerization state of actin regulates the amount of nuclear ezrin. Interestingly, the amino-terminal half of ezrin, which contains mainly the FERM domain, localizes predominantly to the nucleus (Kaul et al, 1999; Stokowski & Cox, 2000). This would suggest that the carboxy-terminus of ERMs plays a role in the regulation of their nuclear localization; however, it is also possible that the C-terminal domain might simply retain the protein in the cytoplasm through binding to F-actin. Phosphorylation of ezrin at Y354 was found to be necessary for nuclear localization (Di Cristofano et al, 2010), but this has not been confirmed in the case of the other paralogs. The NES and exportin of ERM proteins are not yet known, but human ezrin was identified in a proteomics screen as a cargo for exportin-1 (CRM1), a highly conserved, RanGTPase-driven exportin (Kırlı et al, 2015).

Invertebrates have only one representative of the family. It has been shown that the only ERM protein of *Drosophila*, moesin, is also present in the nucleus, where it participates in mRNA export (Kristó et al, 2017) and regulates gene expression (Bajusz et al, 2021). The physical interaction between Drosophila moesin and PCID2, an mRNA export factor, has also been demonstrated (Kristó et al, 2017). Recently, the FERM domains of the human ERMs and Merlin were screened against intrinsically disordered regions of the human proteome, and a great number of novel ligands have been validated, among them nuclear proteins such as NOP53, HIF1A, TBX4, and SETD2 (Ali et al, 2023). These findings suggest that the nuclear activity of ERM proteins is much more significant than we think today.

## Merlin

Merlin (also known as NF2) is a membrane–cytoskeleton scaffolding protein; it links actin filaments to a number of transmembrane and endosomal proteins (McClatchey & Fehon, 2009; Michie et al, 2019). The presence of Merlin in the nucleus and nucleolus has been reported already in the 90s (Obremski et al, 1998; Scoles et al, 1998),

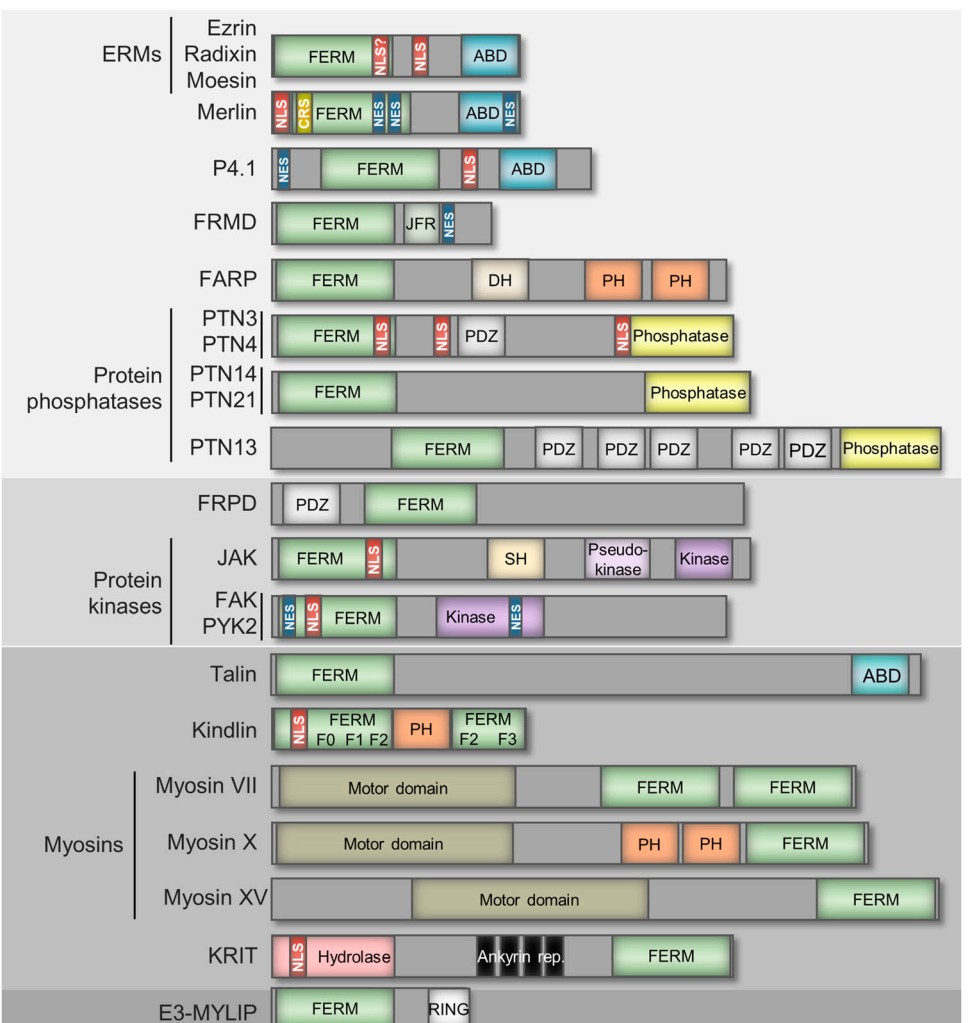

**Figure 2. Domain structure of FERM domain–containing proteins.**
Proteins are grouped according to the phylogenetic relations of the FERM domains (Ali & Khan, 2014). Only the most relevant domains are shown; size is not for scale. Structures are shown in the N- to C-terminal direction. Known NLS, NES, and cytoplasmic retention motifs are highlighted with red, dark blue, and yellow rectangles, respectively.

later showing that it is distributed diffusely within the nucleoplasm (Kressel & Schmucker, 2002). Similar to ERM proteins, Merlin's nuclear localization is controlled by cell density. Through the dephosphorylation of Merlin at S518, high cell density induces the formation of a closed-conformation protein capable of nuclear localization (Li et al, 2010; Hikasa et al, 2016). The amount of Merlin in the nucleus is regulated by NLS and NES, as well as cytoplasmic retention (CRS) motifs. The NLS was identified in human Merlin as a non-canonical sequence at position 24–27 (VRIV), adjacent to the N-terminus of the FERM domain (Li et al, 2014). However, this motif seems to be present in vertebrates only and it is not fully conserved among them either (Fig 4B). The canonical leucine-rich NES sequence ($Lx_{1-3}Lx_{2-3}LxL$) has been mapped to the C-terminal actin-binding domain between amino acids 535–551 in exon 15 of the human Merlin protein (Kressel & Schmucker 2002). Later, two additional NES motifs were identified in the FERM domain of human Merlin, $L_{232}$LLGVDALGLHI and $V_{292}$NKLILQLCI, and the nuclear localization was found sensitive to leptomycin B (LMB) treatment, suggesting that the protein is subject to exportin-1–dependent nuclear export (Furukawa et al, 2017). All three NESs are conserved

from insects to humans, indicating the importance of nuclear export of Merlin (Furukawa et al, 2017). Besides the NLS and NES, a third motif also regulates the cellular distribution of Merlin through cytoplasmic retention ensured by a 25–amino-acid-long section in the F1 lobe of the FERM domain (Kressel & Schmucker, 2002). Multiple sequence alignment uncovers that this CRS motif is shorter than proposed earlier, and is not only present in the FERM domain of human Merlin and protein 4.1 proteins, but it is also conserved across the Merlin, protein 4.1, and ERM families (Fig 4C). Although the protein binding the CRS motif has not been identified yet, it has been shown that under normal cell culture conditions, the nuclear amount of Merlin is determined by its import, but not the export, rate (Kressel & Schmucker, 2002), and the protein undergoes nucleo-cytoplasmic shuttling in a cell cycle– and anchorage-dependent manner (Muranen et al, 2005; Furukawa et al, 2017).

In the nucleus, Merlin binds the E3 ubiquitin ligase CRL4[DCAF1] critical for cell growth. The binding inhibits the interaction of CRL4[DCAF1] with Lats1; therefore, Merlin indirectly promotes the phosphorylation of the transcription coactivator YAP in the nucleus, which in turn induces proliferation arrest (Li et al, 2010; Li et al,

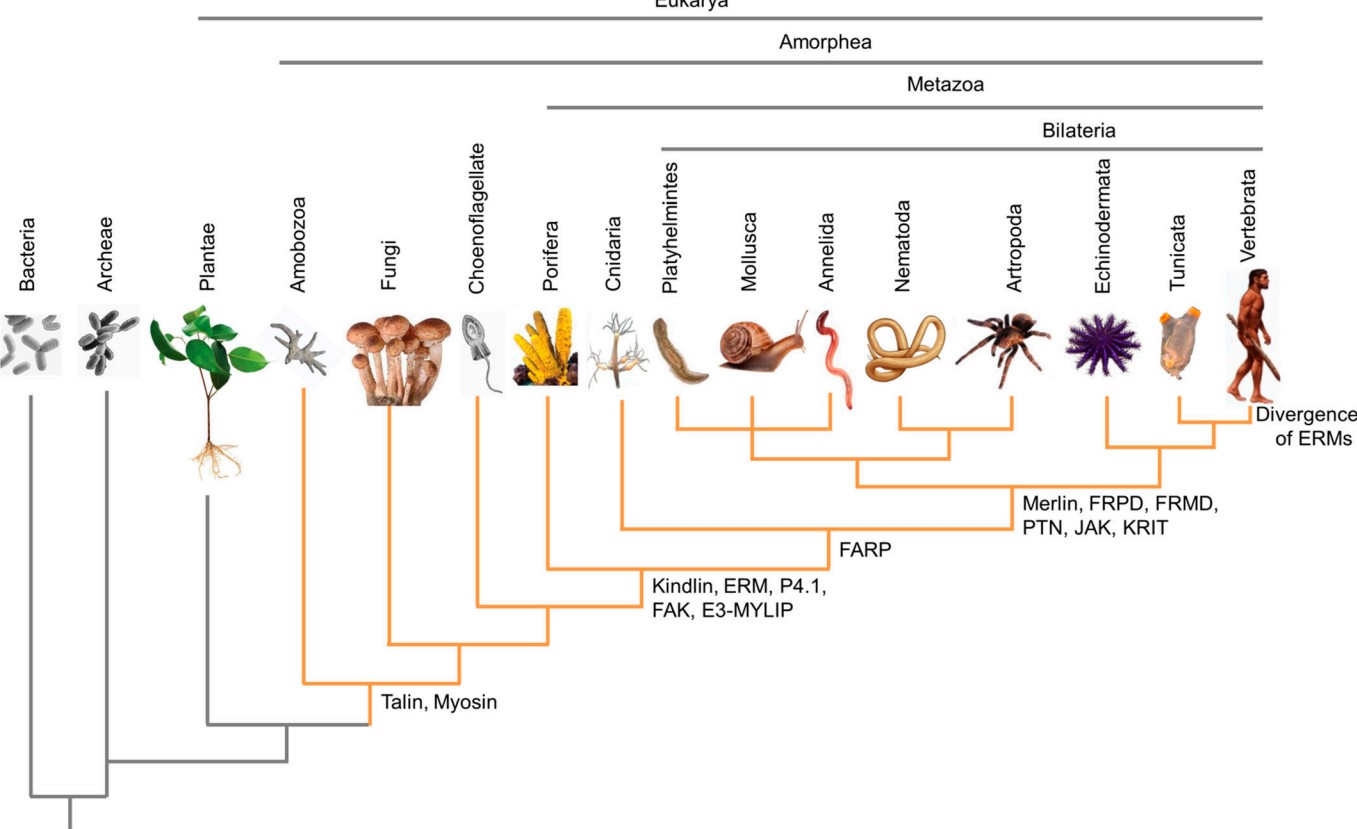

**Figure 3. Evolutionary history of FERM domain–containing proteins.**
A simplified dendrogram below the species images represents evolutionary relationships and shows the origin of selected FERM domain–containing proteins. Major taxonomic groups are shown above the species images. (The former name of Amorphea was Unikonta.) The figure is based on the publication of Ali and Khan (2014).

2014). CRL4^DCAF1 binds to the groove of the F3 lobe of the FERM domain in Merlin (Li et al, 2014), but the domains and residues critical for this binding are only present in proteins from vertebrates, suggesting that the interaction has evolved in chordates to restrain the activity of Hippo signaling (Li et al, 2014). Interestingly, a Merlin mutant lacking all three NESs preserved the interaction with YAP but failed to suppress the nuclear localization of YAP. The most plausible explanation for this might be that the nuclear export of Merlin is involved in the removal of YAP from the nucleus to suppress its activity. Thus, one of the nuclear functions of Merlin may be the regulation of the nuclear transport of other proteins. In the nucleus of endothelial cells, the scaffolding protein NHERF1/EBP50 was also found to bind the FERM domain of Merlin (Finnerty et al, 2004). The interaction retains Merlin in the nucleus and is believed to contribute to the regulation of YAP activity by Merlin (Boratkó et al, 2017).

It was shown that cell–cell contact inhibition triggers Merlin dephosphorylation at S518 (Chen et al, 2011), which promotes the interaction between Merlin and Lin28B, a nucleolar pre-miRNA–binding protein (Hikasa et al, 2016). The sequestration of Lin28B enables the maturation of pri-let-7 miRNAs; therefore, Merlin inhibits cell growth. Another nucleolar protein, PICT-1 (also known as NOP53), has also been reported to bind Merlin, and this interaction modulates the inhibitory effect of PICT-1 on the cell cycle and proliferation (Chen et al, 2011).

## Protein 4.1

Protein 4.1 is essential for regulating cell membrane physical properties of mechanical stability and deformability through stabilizing the interaction between spectrin and actin (Krauss et al, 2003). The first report that 4.1R, one of the four paralogous mammalian 4.1 proteins, shows strong nuclear localization was published early (Correas, 1991), and 4.1N was also found to translocate to the nucleus after growth factor treatment (Ye et al, 1999). Multiple laboratories demonstrated that protein 4.1 localizes diffusely in the nucleoplasm except for the nucleoli (de Cárcer et al, 1995; Krauss et al, 1997; Mattagajasingh et al, 1999). The NLS of 4.1R is composed of a cluster of basic amino acids (K648KKRER) located outside the FERM domain, and generated by the joining of exons 13 (first lysine) and 16 (Luque et al, 1998). An N-terminal acidic EED motif was also found to be essential for nuclear import (Gascard et al, 1999). The direct interaction between protein 4.1R and the importin-$\alpha$ (Rch1)/$\beta$ complex was also confirmed, and according to the model, the positively charged KKKRER and the negatively charged EED motifs are close to each other in space, and together, they interact with importin-$\alpha$ (Gascard et al, 1999). Interestingly, the KKKRER sequence exhibits evolutionary conservation in 4.1 proteins from vertebrate species only (Fig 4D), but it is not present in the other 4.1 proteins encoded by the paralogous genes in mammals

**Table 1.  Summary of nuclear localization and function of FERM domain–containing proteins.**

| Protein | NL reported | NL induction | Exportin-1–mediated export | Nuclear interacting partner | Nuclear function |
|---|---|---|---|---|---|
| ERM | + | High cell density, HS, GH, inhibition of mRNA export | | PCID2, NOP53, HIF1A, TBX4, SETD2 | mRNA export, transcription |
| Merlin | + | High cell density | + | CRL4$^{DCAF1}$, NHERF1, Lin28B, PICT-1 | Regulation of nuclear transport and the Hippo pathway, inhibition of cell growth and proliferation. |
| P4.1 | + | GH (NGF) | + | Emerin, lamin A, NuMA, PIKE, SC35, U2AF$^{35}$ | mRNA splicing, nuclear architecture maintenance |
| FRMD | + | | + | – | Regulation of c-Met function |
| FARP | + | | | | |
| PTN3/ 4, 13 | + | | | | |
| PTN14/ 21 | + | Low cell density | | | Dephosphorylation and retention of nuclear YAP |
| FRPD | + | | | | |
| JAK | + | | | STAT3, STAT1, IFNGR1, NF1-C2, RUSH-1α | Nuclear retention of STAT1, gene expression regulation through H3 histone phosphorylation, X chromosome inactivation, nuclear reprogramming. |
| FAK | + | Induction of apoptosis, oxidative stress, shear stress, RA treatment, cell de-adhesion, elevated Ca2+ level | + | P53, Mdm2, GATA4, CHIPS, MBD2, IL-33, ST2 | Acting as a scaffold to stabilize complexes regulating the transcription of immunosuppressive chemokine and cytokine genes, direct regulation of transcription factors |
| Kindlin | + | | | Chromatin | Regulation of DE-cadherin expression? |
| Talin | + | | | Active β-catenin | Enhancement of β-catenin transcription factor activity and *Axin2* gene expression |
| MyoVII, XV | | | | | |
| MyoX | + | | + | | |
| KRIT | + | | + | | |
| E3-MYLIP | | | | | |

NL, nuclear localization; HS, heat stress; GH, growth hormone; RA, retinoic acid.

and homologous proteins of invertebrates. The EED amino acid triplet is located in the F1 lobe of the FERM domain, and it has been implicated in the direct interaction with the cytoplasmic domain of the plasma membrane transporter NHE1 (Na+/H+ exchanger isoform 1) (Hideki Wakui et al, 2013). In addition, it is also part of the conserved CRS described in the case of human Merlin (Fig 4C). The possibility of a functional CRS around EED is supported by the observation that exon 5 encoding the CRS motif inhibits the nuclear import of protein 4.1. Exon 5 is an alternative exon, its skipping by splicing contributes to nuclear translocation of protein 4.1, and it is also able to confer cytoplasmic localization to a nuclear reporter (Luque & Correas, 2000). No data are currently available for the structure of the full-length 4.1R protein, but the 3D structure predicted by AlphaFold (Jumper et al, 2021; Varadi et al, 2022), although uncertain because of the disordered region, places the EED

and NLS motifs far apart (Fig 4E). Whether the EED triplet supports or inhibits nuclear import needs to be clarified in the future. Interestingly, the N-terminal unstructured domain of 209 amino acids in the 135 kD molecular weight isoform of protein 4.1 was also found to hinder nuclear entry (Luque et al, 1999; Nunomura et al, 2011). The nuclear export of protein 4.1R is controlled by the conserved, hydrophobic NES sequence L26LKRVCEHLNLL, recognized by exportin-1 (Luque et al, 2003).

The nuclear binding partners of protein 4.1 identified so far point to a role in maintaining nuclear architecture and mRNA splicing. 4.1R was identified in a mass spectrometry analysis of mammalian nuclear envelope proteins (Schirmer et al, 2003), and the inner nuclear membrane–associated proteins, emerin and lamin A, coimmunoprecipitated with 4.1R (Meyer et al, 2011). The presence of protein 4.1 on the nuclear pores (Krauss et al, 1997) and on the

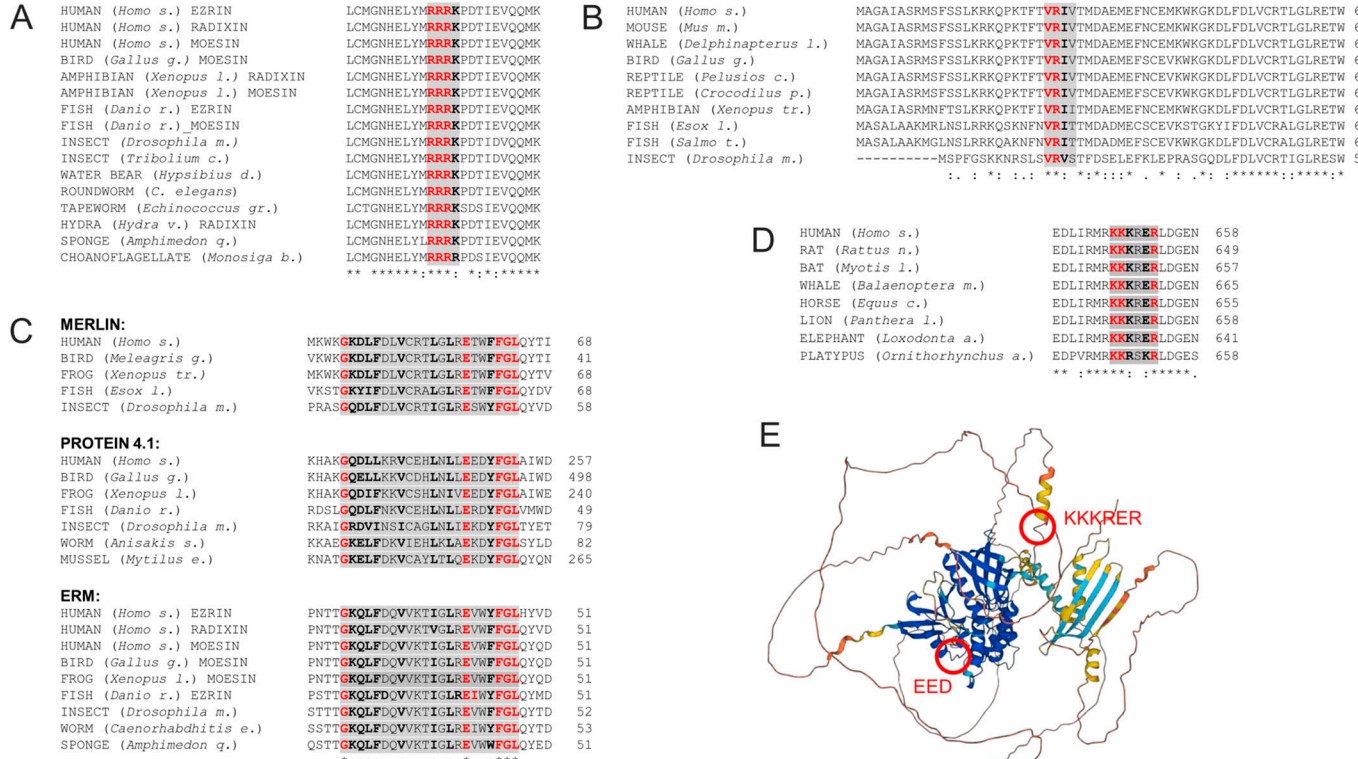

**Figure 4. Evolutionary conservation of verified NLS and cytoplasmic retention motifs regulating the nuclear import of FERM domain proteins.**
**(A)** Conservation of the NLS (highlighted with gray background) predicted in the FERM domain of ERMs. **(B)** Conservation of human Merlin NLS (gray background) and the N-terminal end of the FERM domain. **(C)** Conservation of the cytoplasmic retention (gray background) identified in the FERM domain of human Merlin extends to the protein 4.1 and ERM families. **(D)** Conservation of the NLS (gray background) of vertebrate protein 4.1R proteins. **(E)** AlphaFold structure prediction of full-length protein 4.1R based on the sequence of UniProt ID P11171. Under the protein sequences, asterisks (*) indicate positions, which have fully conserved residues (also indicated in red and bold letters); colon (:) indicates conservation between amino acids of strongly similar properties (marked in bold letters); subscript period (.) indicates conservation between residues of weakly similar properties; and no symbol indicates no conservation.

intranuclear filaments (Kiseleva et al, 2004) was also reported. These results and the finding that depletion of 4.1R affects nuclear structures together suggest that the protein acts as a linker or adapter at nodes vital for interconnections between nucleoplasmic substructures, nuclear envelope components, and the nucleo-cytoskeletal interface (Meyer et al, 2011). The last 64 amino acids at the C-terminus of protein 4.1R interact with the structural protein NuMA, present in the nucleoplasm during interphase (Mattagajasingh et al, 1999). The interaction is necessary for P4.1 nuclear localization (Mattagajasingh et al, 2009) and nuclear assembly (Krauss et al, 2002), but the exact molecular mechanisms behind these activities of P4.1 are presently unknown. Treatment with NGF hormone induces the translocation of 4.1N to the nucleus by an as yet unknown mechanism, where it associates with NuMA and therefore prevents its role in nuclear assembly at the end of mitosis (Ye et al, 1999). Phosphorylation of protein 4.1 at T60 and S679 by CDK1 kinase enhances this binding (Huang et al, 2005; Treviño et al, 2010). These findings demonstrate further that the regulation of the nuclear import of mammalian protein 4.1 proteins is very complex; it involves multiple alternative splicing and phosphorylation events.

The colocalization of protein 4.1 and actin was detected in mammalian cell nuclei (Krauss et al, 2003), which also points to a structural function, although considering the manifold and not primarily structural functions of nuclear actin (Kloc et al, 2021); this observation suggests that the role of P4.1 in the nucleus might not be limited to structural tasks. The idea is supported by the result according to which after NGF-induced nuclear translocation, rat protein 4.1N binds to nuclear GTPase PI 3-kinase enhancer (PIKE) and prevents its interaction with phosphoinositide 3-kinase (PI3K). This leads to the decline of the nuclear PI3K activity up-regulated by NGF (Ye et al, 2000). In mammalian and Xenopus nuclei, protein 4.1R localizes to the mRNA splicing factors, the nuclear speckles (Correas, 1991; de Cárcer et al, 1995; Krauss et al, 1997). The direct interaction between protein 4.1 and the pre-mRNA splicing factors SC35 (Lallena & Correas, 1997) and U2AF35 (Lallena et al, 1998) was demonstrated by coimmunoprecipitation. These results reveal that P4.1 is an active participant in pre-mRNA splicing, and demonstrate that its role in the nucleus is more than just maintaining structure.

## FRMD

The human genome harbors seven genes that encode seven distinct proteins, which are all made up of an N-terminal FERM domain and a subsequent disordered region of 2–300 amino acids (Fig 2). These proteins have been named FERM domain–containing proteins (FRMDs), and the best known of them is the human FRMD6,

also known as Willin. FRMD6 is a multifunctional protein, an upstream regulator of Hippo signaling that has recently been shown to modulate actin cytoskeleton dynamics (Chen et al, 2021).

Nuclear localization of FRMD6 was observed in squamous cell carcinomas of the head, neck, and upper aerodigestive tract (Moleirinho et al, 2013). The NLS protein motif required for nuclear transport of FRMDs is not yet known, but the nuclear translocation of FRMD6 is regulated by T28 phosphorylation and association with 14-3-3 proteins. This strongly suggests that the nuclear import of FRMD6 is controlled, at least partly, by cytoplasmic retention (Meng et al, 2015). A functional NES motif (L$_{344}$SDVSKQVEDLRL) was identified outside the FERM domain of human FRMD7, and LMB treatment induced the nuclear accumulation of the protein, indicating exportin-1–mediated nuclear export and further confirming that the nuclear localization of FRMD proteins is regulated by their transport (Watkins et al, 2013). In the nuclei of glioblastoma cells, FRMD6 colocalizes with c-Met, a growth factor receptor tyrosine kinase, suggesting that FRMD6 may regulate c-Met functions such as calcium signaling in the nucleus (Xu et al, 2016).

### FARP

The FERM, ARHGEF (RhoGEF), and two PH domain–containing proteins (FARPs) function as guanine nucleotide exchange factors for RAC1 RhoGTPase (Koyano et al, 1997). Among other things, FARPs regulate the structure of the actin cytoskeleton, through which they play a role in the formation of synapses (Kubo et al, 2002). Human full-length FARP1 localizes mainly in the cytoplasm; however, N-terminally truncated FARP1 becomes nuclear, suggesting that the FERM domain may regulate the subcellular localization of FARPs (Kubo et al, 2002). Human FARP2 was identified in a proteomics screen as a cargo for exportin-1 (Kırlı et al, 2015), suggesting regulated nuclear localization for FARPs. However, the significance of their nuclear localization and their nuclear interacting partners or function is still unknown.

### Protein phosphatases

Protein tyrosine phosphatases (PTPs) control the reversible phosphorylation of tyrosine residues and are key regulators of essential signal transduction pathways (Young et al, 2021). PTPs are primarily cytosolic, but their activity is also needed in the nucleus, where dephosphorylation of members of the STAT family is important for terminating STAT signaling (Buck et al, 2022). Some of the non-transmembrane types of PTPs acquired the FERM domain during the course of evolution. These phosphatases are PTN3 (PTP-H1), PTN4 (MEG1), PTN13 (PTP-BAS/PTP-BL), PTN14 (PTPD2), and PTN21 (PTPD1). They are primarily cytoskeleton-associated, but PTN3 (Hsu et al, 2007), PTN4 (Szczałuba et al, 2018), PTN13 (Cuppen et al, 2000; Nakahira et al, 2007), and PTN14 (Wadham et al, 2000) were detected in the nucleus. Low cell density induces nuclear translocation of PTN14 (Wadham et al, 2000), and in PTN3, three functional NLS motifs, K264RKK, R389KPR, and R655KKP, were identified (Hsu et al, 2007). The K264RKK sequence resides at the C-terminal end of the FERM domain, whereas R655KKP is part of the phosphatase domain (Fig 2). Interestingly, the mutation of these three NLSs inhibits nuclear localization only if the FERM domain is

removed at the same time, suggesting that the FERM domain retains the protein in the cytoplasm.

STAT4 and STAT6 phosphorylation was increased in the nuclei of PTN13-deficient CD4$^+$ T cells, and the inhibitory effect of PTN13 on STAT phosphorylation was evident in the nuclear fraction (Nakahira et al, 2007), which suggested that PTN13 can function as a protein phosphatase also in the nucleus. Accordingly, it was later demonstrated that nuclear YAP is a direct target of PTN14, and knockdown of PTPN14 induces the nuclear retention of YAP (Liu et al, 2013). These results indicate that PTPN14 in the nucleus suppresses the transcriptional coactivator activity of YAP and assists in the removal of inactive nuclear YAP, thereby inhibiting YAP-dependent cell migration.

### FRPD

FERM and PDZ domain–containing proteins (FRPDs or FRMPDs) are represented by four paralogs in mammals, FRPD1-4. They are implicated in a wide range of morphogenic and signaling functions through maintaining and modulating synaptic transmission (Lee et al, 2008; Ueno et al, 2018). The localization of FRPD4 (also known as Preso1) in the nuclei of neuroblasts immediately after asymmetric cell division was mentioned a decade ago (Lee et al, 2014), but at present, virtually nothing is known about the possible nuclear transport and activity of FRPDs.

### JAK

The Janus kinase (JAK) family consists of four mammalian members, JAK1-3 and tyrosine kinase 2 (TYK2). They are non-receptor–type tyrosine kinases, and are key players in a large number of signaling pathways. Several studies confirmed the nuclear localization of both Jak1 and Jak2 (Zouein et al, 2011). In addition, in rat (Ram & Waxman, 1997) and human (Hellgren et al, 1999) hepatocytes, constitutive nuclear localization of JAKs was observed, and JAK2 appeared to colocalize with chromosomes (Ito et al, 2004). TYK2 was also detected in the nucleus of human fibrosarcoma cells, with the exclusion of nucleoli (Ragimbeau et al, 2001).

The amino acids K$_{342}$RKK in the F3 lobe of the FERM domain form a classical NLS in human JAK1 (Fig 2). The motif can be recognized by the importin-α isoforms α4, α5, and α7, and JAK1 is constitutively imported into the nucleus regardless of its activation status (Zhu et al, 2017). Accordingly, unphosphorylated JAK2 was found to be constitutively present in the cell nucleus and was capable of undergoing activation there (Noon-Song et al, 2011). In the case of TYK2, nuclear localization requires an Arg-rich NLS located around residues 219–240 within the FERM domain. Although the motif (R$_{219}$RHIRQHSALTRLRLR) resembles a bipartite NLS, further experiments suggested that it is required but not sufficient for nuclear import, and that other regions in Tyk2 contribute to its activity (Ragimbeau et al, 2001).

JAK1 was found pivotal for the viability of human B cells (Zhu et al, 2017) indicating essential nuclear functions for JAKs. In the nucleus, JAK1 and JAK2 directly interact with STAT3 (Ram & Waxman, 1997), STAT1 (Mowen & David, 2000), and IFNGR1, the ligand-binding chain of the gamma interferon receptor (Noon-Song et al, 2011). The direct phosphorylation of nuclear STAT1 by JAK1 inhibits the NES of STAT1

and therefore retains the protein in the nucleus (Mowen & David 2000). The direct involvement of JAK in gene expression regulation by chromatin phosphorylation was first discovered in *Drosophila* (Shi et al, 2006). Later, nuclear JAK2 was found to phosphorylate H3 histone at Y41, which modification excludes HP1α from chromatin and promotes gene expression (Dawson et al, 2009). The same epigenetic modification was also described for JAK2 (Rui et al, 2010; Griffiths et al, 2011), and its physiological significance in embryonic stem (ES) cell self-renewal has also been demonstrated (Griffiths et al, 2011). The phosphorylation of H3 histone by JAKs was found at the promoter of the *IRF1* (Noon-Song et al, 2011), *nanog* (Griffiths et al, 2011), and *myc* (Rui et al, 2016) genes. Nuclear JAK2 also phosphorylates the transcription factor NF1-C2, thereby preventing its proteasomal degradation (Nilsson et al, 2006), and the rabbit HLTF orthologue RUSH-1α transcription factor (Helmer et al, 2011). JAK1 was also reported to be a major regulator of nuclear reprogramming induced by endoplasmic reticulum stress (Sims & Meares, 2019). Recently, JAKs have been suggested to play a role also in the maintenance of X chromosome inactivation, which uncovers a potential novel aspect of nuclear function for JAK proteins (Lee et al, 2020).

## FAK

Focal adhesion kinases (FAKs) are non-receptor tyrosine kinases that play a crucial role in cellular adhesion, migration, and proliferation through the binding of more than 50 proteins (Zhou et al, 2019). FAK and proline-rich tyrosine kinase 2 (Pyk2) (also known as FAK2 or PTK2B), the other member of the FAK family, exhibit kinase-dependent and kinase-independent functions as a scaffold recruiting various proteins. It also became evident very soon that FAKs are an active component of the nucleus (see the reviews: Lim, 2013; Pomella et al, 2022). The clear confirmation of the presence of FAK in the nucleus and its possible functioning there comes from the experiments in which the low steady-state levels of nuclear FAK greatly increase to stress such as the induction of apoptosis, oxidative stress, shear stress, retinoic acid treatment, inhibition of nuclear export, or cell de-adhesion (Lim et al, 2008; Ossovskaya et al, 2008; Luo et al, 2009; Ahn & Park, 2010; Lim, 2013; Sanchez et al, 2016). The surface of the F2 lobe of the FERM domain in both FAK and Pyk2 harbors a functional NLS. The amino acids KK190-191 and K216K218R221K222 in the human FAK protein are essential components of the motif (Lim et al, 2008). SUMOylation of the FERM domain at K152 induces nuclear accumulation of FAK (Kadaré et al, 2003), but this modification is not essential for nuclear translocation (Lim et al, 2008). In human FAK, the L90RSEEVHWLHVDM sequence in the F1 lobe of the FERM domain is a functional NES, although the motif is conserved in vertebrate FAKs and Pyk2 only (Ossovskaya et al, 2008). The kinase domain of FAK is also involved in the nuclear export; the $L_{518}$DLASLIL sequence forms a second functional NES motif (Ossovskaya et al, 2008; Lim, 2013) (Fig 2), which is recognized by exportin-1 (Jones & Stewart, 2004; Lim et al, 2008).

The import mechanism of Pyk2 appears to be significantly different from that of FAK. Elevated Ca2+ concentration is a prerequisite for Pyk2 activation and subsequent nuclear translocation (Faure et al, 2007). Interestingly, the classical NLS located on the surface of the F2 lobe of the FERM domain of FAK and Pyk2 is not sufficient for regulated nuclear accumulation of Pyk2, but an evolutionarily conserved S747PT motif, also known as nuclear translocation signal (NTS), plays an accessory role in the nuclear import by increasing the nuclear transfer rate (Faure et al, 2013). NTS sequences contain an S/T-P-S/T motif, which when phosphorylated binds to importin-7 (Chuderland et al, 2008). NTS sequences have recently been shown to be located in regions proximal to the NLS motif EKRKI(E/R)(K/L/R/S/T) recognized by importin-7 (Panagiotopoulos et al, 2021). Interestingly, the sequence of PYK2 does not contain this binding site of importin-7 or any other canonical NLS site near the NTS, so the exact mechanism of the nuclear import of PYK2 has yet to be explored. The nuclear export of Pyk2 is driven by the NES motif L735QFQV in the 700–841 linker region and is regulated by phosphorylation at S778. However, an additional residue in the 767–793 region is also necessary for nuclear export. Like FAK, the nuclear export of Pyk2 is LMB-sensitive, indicating that exportin-1 transports Pyk2 to the cytoplasm (Faure et al, 2013).

Activated FAK has been shown to interact with a number of nuclear proteins (reviewed in Zhou et al [2019]), and not only acts as a scaffold to stabilize complexes, but it can also directly regulate transcription factors. With its FERM domain, nuclear FAK simultaneously binds p53 and the E3 ubiquitin ligase Mdm2 providing a scaffold for p53 polyubiquitination and degradation (Lim, 2013). Using the same mechanism, FAK also promotes inflammation by concurrently interacting in the nucleus with the transcription factor GATA4 and the E3 ubiquitin ligase protein CHIPS (Lim et al, 2012). The scaffolding function of FAK is also used to regulate gene expression. It forms a complex with MBD2 that recruits the NURD complex to methylated CpG promoter sites, and as a result, they promote the dissociation of HDAC1 from an MBD2-HDAC1 complex, which in turn inhibits myogenin transcription (Mei & Xiong, 2010).

In addition to scaffolding functions, FAK can bind and directly regulate the activity of several transcription factors, such as the inflammatory factor IL-33 and its receptor, ST2 (or IL1RL1) (Serrels et al, 2017), as well as the transcription repressor methyltransferase EZH2 (Gnani et al, 2017), the Runt-related transcription factor 1 (RunX1), which is involved in the generation of hematopoietic stem cells (Canel et al, 2017), and the TATA-binding protein–associated factor of the TFIID preinitiation complex, TAF9 (Serrels et al, 2015). Through the interaction with TAF9 (Serrels et al, 2015), FAK plays a role in the expression of immunosuppressive chemokines and cytokines Ccl5 and TGFb2. Active FAK was detected also in the nucleolus, where it protects nucleostemin, a nucleolar GTPase that safeguards mitotic stem/progenitor cells from DNA damage in the S-phase, from proteasomal degradation (Tancioni et al, 2015), thereby promoting its function in ribosomal biogenesis and proliferation.

## Talin

Talins are high-molecular-weight cytoskeletal adapter proteins that are primarily known for their role in linking integrin receptors to the actin cytoskeleton at focal adhesion sites. Talin contains an N-terminal FERM domain that binds and regulates the conformation of the integrin receptor and induces intracellular force sensing, and a C-terminal F-actin–binding domain (Fig 2). In human

epithelial cells, talin-1 was found recently in the nucleus, where it interacts with the chromatin and localizes primarily to the nucleolus (Da Silva et al, 2022 Preprint). In Drosophila, the talin homologue (rhea) regulates DE-cadherin at the transcriptional level; however, at present, it seems more likely that talin acts in the cytoplasm to control the activity of a transcriptional factor rather than being directly involved in gene expression in the nucleus (Bécam et al, 2005).

### Kindlin

Kindlins are cytoskeletal scaffolds or adapters, and are essential in cell-to-cell crosstalk via cell–cell contacts and integrin-mediated cell adhesion. The kindlin protein family consists of three conserved protein homologues in mammals, kindlin-1, kindlin-2, and kindlin-3 (Rognoni et al, 2016). The F1 lobe of their FERM domain is preceded by an N-terminal F0 subdomain, and the F2 lobe is interrupted by a PH domain (Plow & Qin, 2019) (Fig 2). Because a significant proportion of focal adhesion proteins, including migfilin, a kindlin-binding protein (Wu, 2005), are also active in the nucleus (Hervy et al, 2006; Haage & Dhasarathy, 2023), it is not surprising that kindlins were also found in the nucleus. In human keratinocytes, kindlin-1 was found in the nucleus (Lai-Cheong et al, 2008), and kindlin-2 was identified almost exclusively in the nuclei of smooth muscle cells (Kato et al, 2004) and observed in the nuclei of breast cancer cells (Yu et al, 2012). A predicted NLS between amino acids 55–72 was identified in kindlin-2; however, the motif is exclusively present in kindlin-2 (Ussar et al, 2006). The NES motif of kindlins is yet unknown, but both human kindlin-1 and kindlin-2 were found as a positive hit in a global proteomics screen identifying exportin-1 binding partners (Kırlı et al, 2015). It has been shown that kindlin-2 specifically interacts with the active form of β-catenin in the nucleus of several human cells, forming a tripartite complex with β-catenin and TCF4. The interaction selectively strengthens the occupancy of β-catenin on the Wnt target gene Axin2 and therefore promotes its expression (Yu et al, 2012).

### Myosins

Until now, a large number of myosin superfamily members have been demonstrated to localize to the nucleus and have roles in transcription, DNA repair, chromatin dynamics, intranuclear transport, and viral infections (Cook et al, 2020; Maly & Hofmann, 2020; Venit et al, 2020). FERM domains are found in the tails of evolutionarily distant myosins, MyoVII, MyoX, and MyoXV (Planelles-Herrero et al, 2016), and the domain interacts with adhesion and signaling receptors, and actin-binding proteins (Boëda et al, 2002; Zhang et al, 2004; Liu et al, 2008). Among the myosins containing the FERM domain, only myosin-X of vertebrates has so far been confirmed to be present in the nuclei of Xenopus epithelial cells (Woolner et al, 2008), but the significance of this nuclear presence is not known. To date, no nuclear transport motif was identified for the FERM domain–containing myosins, although human myosin-X was found as an exportin-1 binder candidate (Kırlı et al, 2015), indicating regulated nuclear transport, but this result remains to be validated. Accordingly, the exploration of the nuclear interacting partners and functions of MyoVII, MyoX, and MyoXV is also the task of future research.

### KRIT

Krev interaction trapped protein-1 (KRIT1) is a scaffolding protein that plays a critical role in vascular morphogenesis and homeostasis (Fisher & Boggon, 2014). KRIT1 contains a C-terminal FERM domain (Fig 2) that interacts with the small GTPase Krev-1 (Rap1) (Serebriiskii et al, 1997), and a Nudix domain that contains a functional NLS (residues 46–51) (Zhang et al, 2000; Draheim et al, 2017). LMB treatment causes the nuclear accumulation of KRIT1, indicating that exportin-1 is responsible for the removal of the protein from the nucleus (Zhang et al, 2007). The nucleo-cytoplasmic shuttling of KRIT1 was observed (Zhang et al, 2007), and ICAP1α was found to stabilize KRIT1 and drive it into the nucleus (Draheim et al, 2017). Available data suggest that the key role of KRIT1-ICAP1α interaction and nuclear import is to prevent KRIT1 from proteasomal degradation, and the nuclear retention of ICAP1α by KRIT1 is a tool to modulate the cytoplasmic activity of ICAP1α (Draheim et al, 2017). This result demonstrates the dual role of KRIT1 as a cytoplasmic and a nuclear protein, but its exact function in the nucleus remains at present completely unknown.

### E3-MYLIP

E3 ubiquitin-protein ligase (E3-MYLIP) (also known as IDOL, BZF1, or MIR) mediates ubiquitination and subsequent proteasomal degradation of myosin regulatory light chain (Olsson et al, 1999) and LDL receptors (Zelcer et al, 2009; Hong et al, 2010; Calkin et al, 2011). The nuclear localization of the protein has not yet been reported.

## Concluding Remarks

The FERM domain as a switchable interaction HUB has been applied for many activities during evolution and makes FDCPs very versatile, so it is not surprising that their functions are not limited to the cytoskeleton alone. Based on the data collected here, it is clear that FDCPs are present in the cell nucleus; in fact, there is obviously a strictly regulated amount of them in the nucleus, which is in dynamic equilibrium with their cytoplasmic pool. The effects that induce their nuclear localization seem to be diverse (Table 1), but their nuclear transport motifs and transportins, in particular, in the case of their export, are still practically completely unknown. Similarly, little is known today about the regulation of their nuclear transport, although based on what has been reported so far, phosphorylation seems to be the most common mode of regulation. Phosphorylation of ERM, protein 4.1, and FRMD6 proteins, and dephosphorylation of Merlin and JAK2 regulate their import, whereas phosphorylation of FAK is needed for its exit from the nucleus. Despite the fact that the FERM domain is highly conserved and carries the motifs necessary for nuclear transport in many FDCPs, the functional NLS and NES motifs often reside outside the FERM domain. However, based on the still rather limited amount of data available in the literature, we can perhaps say that their

nuclear import and export mechanisms are regulated in a diverse manner. This is probably related to the fact that like their functions, the effects that trigger their nuclear translocation are also very diverse. Therefore, obviously one of the important tasks of the future will be to investigate the causes, routes, and mechanisms of the nuclear transport of FERM domain proteins.

The presence of FDCPs in the nucleus may simply be a mechanism that controls and/or limits their availability in the cytoplasm, and the nucleus might serve as a reservoir for inactive proteins. Although this possibility cannot be completely ruled out, the data presented here demonstrate that at least some of them are active components of the nucleus and perform important functions there. Because of their known cytoskeletal functions, they were first described as responsible for nuclear architecture. However, it seems that in the nucleus, their tasks are much less restricted to forming and maintaining structure. The best example of this is ERM proteins, protein 4.1, and JAKs, which are directly involved in splicing, mRNA export, and gene expression. We do not yet have a comprehensive picture of the nuclear binding partners and thus of the nuclear activity of the FERM domain, but because it is an extremely versatile and complex domain, the answer will certainly be diverse. The details will be provided by the identification of binding partners, which will probably open up a new dimension in the future for each family of FDCPs and for the nuclear function itself.

In the nucleus, the basic components and functions, such as the chromatin, nuclear pores, functional compartments, chromosome territories, are all universal and extremely conserved, so we tend to think of the internal organization and function of the eukaryotic nucleus as universal. In contrast, the variability in the nuclear transport and functions of different FERM domain proteins that appeared at different points in the evolution provides clear evidence that the nucleus of eukaryotes evolved and continues to evolve as continuously as the whole cell or even the multicellular organisms made up of it.

Research into the nuclear activity of cytoskeletal proteins, such as FDCPs, is obviously hindered by the fact that because of their often essential cytoplasmic activity, the functional separation of the two compartments in their case is a great challenge. However, the technological development of recent decades, the latest nanoscopic and molecular biological procedures, can give new impetus to these studies, and we can finally understand the significance of the observations from 20 to 30 years ago and the molecular mechanisms behind them. Because nuclear FDCPs represent a link between the cytoplasm and the nucleus, we are convinced that the extension of our knowledge to the nuclear activity of FDCPs will be significant from the point of view of the functioning not only of FDCPs or the nucleus, but also of the eukaryotic cell as a whole.

# Acknowledgements

This work was supported by NKFIH (Hungarian National Research, Development and Innovation Office) through the National Laboratory for Biotechnology program, grant 2022-2.1.1-NL-2022-00008 (to P Vilmos).

## Author Contributions

P Borkúti: conceptualization, formal analysis, and writing—original draft, review, and editing.
I Kristó: data curation and writing—original draft, review, and editing.
A Szabó: visualization and writing—review and editing.
Z Kovács: visualization and writing—review and editing.
P Vilmos: conceptualization, supervision, funding acquisition, project administration, and writing—original draft, review, and editing.

## Conflict of Interest Statement

The authors declare that they have no conflict of interest.

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
