## [Reviewer comments · Life Science Alliance]

Life Science Alliance

FERM domain-containing proteins are active components of the cell nucleus

Peter Borkuti, Ildiko Kristo, Aniko Szabo, Zoltan Kovacs, and Peter Vilmos

DOI: <https://doi.org/10.26508/lsa.202302489>

Corresponding author(s): Peter Vilmos, HUN-REN Biological Research Centre Szeged

Review Timeline:	Submission Date:	2023-11-18
	Editorial Decision:	2023-12-27
	Revision Received:	2024-01-16
	Editorial Decision:	2024-01-19
	Revision Received:	2024-01-20
	Accepted:	2024-01-22

Transaction Report:

December 27, 2023

Re: Life Science Alliance manuscript #LSA-2023-02489

Dr. Peter Vilmos
HUN-REN Biological Research Centre Szeged
Institute of Genetics
62. Temesvari krt.
Szeged 6726
Hungary

Dear Dr. Vilmos,

Thank you for submitting your manuscript entitled "The nuclear localization and functions of FERM domain-containing proteins" to Life Science Alliance. The manuscript was assessed by expert reviewers, whose comments are appended to this letter. We invite you to submit a revised manuscript addressing the Reviewer comments.

Thank you for this interesting contribution to Life Science Alliance. We are looking forward to receiving your revised manuscript.

Sincerely,

B. MANUSCRIPT ORGANIZATION AND FORMATTING:

Reviewer #1 (Comments to the Authors (Required)):

This concise review clearly summarizes the available information about the nuclear localization of FERM domain proteins, their mechanism of import and export (and where relevant, cytosolic retention), and alludes to their possible functions in the nucleus. Overall, this is an informative and well-organized overview, which calls attention to this understudied aspect of the biology of FERM domain proteins. It will help to trigger new research in this area with regard to nucleocytoplasmic shuttling, as well as the regulation and functional significance of this process. The review is definitively a worthwhile contribution to the field.

The authors may wish to consider the following suggestions:

It would be very helpful to generate a table, which summarizes all the key information in a glance.

The table could indicate

- a) the NLS (if it has been identified), and whether it lies (fully or partly) within the FERM domain (direct FERM-mediated import); and whether the molecule also contains alternative NLS;
- b) the proposed import mechanisms (e.g. importin alpha-mediated, , importin-7-mediated, other, unknown);
- c) the identified NES (within and or outside the FERM domain);
- d) the key mechanism underlying altered nuclear accumulation (if identified) and
- e) the proposed or possible functions within the nucleus.

This would make the gaps in our knowledge easily visible and the substantial amount of info easily accessible.

In the introduction, it is stated that the FERM domain is 298 amino acids. Given that many proteins contain FERM domains encoded by different genes (in many species), and that some FERM domains are interrupted with other sequences, it is probably not an absolute rule. Arguably, it would be enough to state that the FERM domain is approx. 30 KDa.

The mechanism and role of Merlin shuttling (especially export) should be described in more details, as this is one of the better-characterized examples of the role of the nuclear localization of FERM proteins. Namely, Furukawa et al reported that a) Merlin contains additional NES sequences beside the previously identified one; and b) these are critical for both Merlin and YAP export from the nucleus. The authors propose that nuclear Merlin binds YAP and facilitates its export via the Merlin NESs. (It chaperons YAP out from the nucleus). Thus, one function of Merlin could be the control of the nuclear transport of other proteins.

Wherever possible, it would be good to provide more information about the regulation of the nuclear transport of FERM proteins. Do various biochemical modifications affect import, export, and/or retention? And if so, how? For example, the authors should mention that the nuclear localization of the ERM proteins and Merlin is regulated by cell density, and they can allude to the proposed mechanisms. Similarly, what is known about the mechanism of NGF-induced nuclear accumulation (entry) of 4.1 or stimulus-induced accumulation of other FERM proteins? It is worth mentioning explicitly if the mechanisms are unknown or perhaps devote a few sentences to this fact in the conclusion.

Reviewer #2 (Comments to the Authors (Required)):

The FERM domain is a protein-module that appears in more than 30 gene-products in the human genome. This domain enables protein-protein and protein-lipid interactions, which determine the localization, and consequently the activity and specificity of the individual FERM domain-containing protein. The FERM domain was thought initially to appear in cytoskeletal proteins with cytoplasmic/membranal localization, although some nuclear activities of such proteins have been reported over the past years. Overall, this is an interesting article that presents the evolution of the FERM domain, which evolved already in bacteria and archaea, and is conserved up to primates. The article also presents a nice review of the structural features of the FERM domain, as well as the evolution of the nuclear localization signal that is responsible for its nuclear localization. Finally, the article covers in detail the properties and regulation of many FERM domain-containing proteins. Nonetheless, I have several minor concerns that should be addressed prior to publication. These are as follows:

1. Although the description of the individual FERM-domain containing proteins is of importance, the information on the role of the FERM domain in some of these proteins is not very focused. Since the review is about FERM domain, it is recommended to make it clearer.
2. Some of the proteins seem to be exported from the nucleus by CRM1. How conserved is it, and how is the NLS/NES equilibrium determined?
3. Similarly, how conserved it's the NTS (as appears in Pyk2), and what are the relation of this sequence with the canonical NLS in this protein.
4. Is there an information on the properties of FERM-deleted proteins?
5. The abstract is somewhat confusing. I had to read it several times to understand it. It is recommended to make it clearer.
6. The terms kinases and phosphates should be replaced to protein kinases and protein phosphatases as other type of kinases and phosphatases also exist but are probably not dependent on FERM domains.

Reviewer #3 (Comments to the Authors (Required)):

In the present manuscript Borkuti et al. aim to review the nuclear localization and functions of FERM domain-containing proteins (FDCPs).

The authors claim that they had provided a comprehensive overview on the "exciting new dimension of the activity of FDCPs, their nuclear transport and function, with special regards to the role of the FERM domain in these processes". In fact, the review lists evolutionary development and conservation of FDCPs, with some hints on the mechanism of nuclear import/export domains, NLS NES etc and their localization in the context of the FERM domain. Insight into nuclear function of most proteins largely remains illusive, or is not described in much detail to really get a comprehensive overview about nuclear function. In most cases, putative interaction partners are listed, not even explained. Hence, the review is not comprehensive without inquiring additional literature. Many of the papers cited are 25-30 years old.

ERM

- Detection in the nucleus
- NLS motives present
- Novel ligands have been validated, among them nuclear proteins
- Function?...not described

Merlin/NF2

- Contains NLS, NES and CRS sequences
- Undergoes nucleo-cytoplasmic shuttling in a cell cycle-dependent manner
- Inhibition of E3 ubiquitin ligase CRLDCAF1: function: restraining activity of Hippo signaling
- Interaction of Merlin with Lin28B (nucleolar RNA-binding protein: function?)
- Binding of Merlin/NF2 to PICT-1: function?

Protein 4.1

- Localizes diffusely in the nucleoplasm
- Nuclear binding partners point to a role in mRNA splicing and maintaining nuclear architecture - which and how?
- 4.1R identified by mass spectrometry in nuclear envelope, associated with emerin and laminA...
- 4.1N associates with NuMA preventing its role in mitosis - which?
- Co-localization of protein 4.1 and actin in nuclei detected...
- In rats binds to PIKE to prevent nuclear PI3K activity
- Localization in mRNA splicing factories, direct interaction with SC35 and U2AF35 (all citations about 25-30 years old!)

FRMD

- Nuclear localization of FRMD6 in diverse cancers
- Co-localization with c-Met in glioblastoma - function?

FARP

- N-terminally truncated FARP becomes nuclear....

Phosphatases

- PTN4, PTN13, PTN14 detected in nucleus
- PTN13 putatively inhibits STAT phosphorylation
- YAP is direct target of PTN14 - function?

FARP

- Nothing really known

JAK

- Nuclear localization of JAK1, JAK2 and TYK2
- JAK2 co-localizes with chromosomes
- Direct interaction of JAK1 and JAK2 with STAT1, STAT3, and IFNGR1 in the nucleus
- Phosphorylation of STAT1 by JAK1 inhibits NES of STAT1
- JAK2 promotes gene expression by phosphorylation of H3
- Potential role of JAKs in maintenance of chromosome X inactivation...

FAK

- Acts as a scaffold to stabilize complexes of p53/MDM2 to promote p53 degradation, or GATA4/CHIPS for the same purpose
- Can directly regulate transcription: forms complexes with MBD2 that recruits the NURD complex to methylated CpG promoter sites, thereby promoting HDAC1 dissociation and inhibition of myogenin transcription
- Direct binding to: IL-33 and its receptor, ST2, TAF9, EZH2, RunX1 - what are the functions of these proteins and their interaction with FAK?
- Detected in the nucleolus - protects Nucleostemin from proteasomal degradation - consequence?

Talin

- Interacts with chromatin - function unknown

Kindlin

- Kindlin2 in the nucleus of different mammalian cells
- Specifically interacts with the active form of β -catenin - function?

Myosins

- MyosinX present in nuclei of Xenopus

KRIT

- Nuclear import through ICAP1 α to prevent its proteasomal degradation in the cytosol
- Function?

In summary, the paper by Borkuti et al. does not present comprehensive information about nuclear functions of FDCPs.

We appreciate the opportunity to submit a revised manuscript. We are grateful for the reviews of our original manuscript and for the constructive and useful comments of the reviewers. We believe that we have addressed all the issues raised and that the contribution is the stronger for it. In addition to the corrections requested by the reviewers:

- We changed the title as the previous one was misleading because it is not yet possible to provide readers a detailed, comprehensive picture of the nuclear transport and function of FDCPs.
- In Figure 4, the symbols and highlights were standardized in the protein sequences, and the description of the figure was completed.
- Crm1 has been renamed Exportin-1 as it is the current official UniProt name.
- The text was adapted to the format requested by Life Science Alliance (running title, summary blurb, reference format, etc.).

Reviewer #1 (Comments to the Authors (Required)):

This concise review clearly summarizes the available information about the nuclear localization of FERM domain proteins, their mechanism of import and export (and where relevant, cytosolic retention), and alludes to their possible functions in the nucleus. Overall, this is an informative and well-organized overview, which calls attention to this understudied aspect of the biology of FERM domain proteins. It will help to trigger new research in this area with regard to nucleocytoplasmic shuttling, as well as the regulation and functional significance of this process. The review is definitively a worthwhile contribution to the field.

The authors may wish to consider the following suggestions:

1. *It would be very helpful to generate a table, which summarizes all the key information in a glance. The table could indicate*
 - a) *the NLS (if it has been identified), and whether it lies (fully or partly) within the FERM domain (direct FERM-mediated import); and whether the molecule also contains alternative NLS;*
 - b) *the proposed import mechanisms (e.g. importin alpha-mediated, , importin-7-mediated, other, unknown);*
 - c) *the identified NES (within and or outside the FERM domain);*
 - d) *the key mechanism underlying altered nuclear accumulation (if identified) and*
 - e) *the proposed or possible functions within the nucleus.*

ANSWER: We also thought about this, but we considered it more illustrative to summarize the data graphically (please see Figure 2). The information about reported nuclear localization, induction of nuclear accumulation, CRM1-mediated export, nuclear interacting partners, and nuclear function are included now in Table 1. However, with the single exception of Merlin, no experimental information is available to date about the import mechanisms, and since we didn't want to fill the table with speculative data, we skipped this from the table.

2. *In the introduction, it is stated that the FERM domain is 298 amino acids. Given that many proteins contain FERM domains encoded by different genes (in many species), and that some FERM domains are interrupted with other sequences, it is probably not an absolute rule. Arguably, it would be enough to state that the FERM domain is approx. 30 KDa.*

ANSWER: This has been corrected.

3. *The mechanism and role of Merlin shuttling (especially export) should be described in more details, as this is one of the better-characterized examples of the role of the nuclear localization of FERM proteins. Namely, Furukawa et al reported that a) Merlin contains additional NES sequences beside the previously identified one; and b) these are critical for both Merlin and YAP export from the nucleus. The authors propose that nuclear Merlin binds YAP and facilitates its export via the Merlin NESs. (It chaperons YAP out from the nucleus). Thus, one function of Merlin could be the control of the nuclear transport of other proteins.*

ANSWER: We are grateful for this comment. This new information has been added to the text as suggested.

4. *Whenever possible, it would be good to provide more information about the regulation of the nuclear transport of FERM proteins. Do various biochemical modifications affect import, export, and/or retention? And if so, how?*

ANSWER: We agree with this suggestion, but as far as we know, very little reliable, experimental data is currently available on this. What is found in the literature (phosphorylation of ERM, Protein 4.1 and FRMD6 proteins, and dephosphorylation of Merlin and JAK2 regulates their import, and phosphorylation of FAK is needed for its export) was mentioned in the manuscript. To be on the safe side, we once again thoroughly reviewed the literature and conducted extensive searches, but found nothing new about the regulatory mechanism raised by the reviewer. However, to emphasize this, we have made this more prominent in the discussion.

5. *For example, the authors should mention that the nuclear localization of the ERM proteins and Merlin is regulated by cell density,*

ANSWER: This was done, and the information has been also included in Table 1.

and they can allude to the proposed mechanisms.

ANSWER: The mechanism is known only in the case of Merlin. Through the dephosphorylation of Merlin at S518, high cell density induces the formation of closed-conformation protein capable of nuclear localization. This has been included in the text.

6. *Similarly, what is known about the mechanism of NGF-induced nuclear accumulation (entry) of 4.1 or stimulus-induced accumulation of other FERM proteins? It is worth mentioning explicitly if the mechanisms are unknown or perhaps devote a few sentences to this fact in the conclusion.*

ANSWER: As mentioned in the manuscript, according to our current knowledge, the regulation of the nuclear import of Protein 4.1 is complex, it involves multiple alternative splicing and NGF-induced phosphorylation events. To our knowledge, the exact mechanisms behind the stimulus-induced accumulation of FDCPs are still practically unknown. This is discussed now in the conclusions, as suggested.

Reviewer #2 (Comments to the Authors (Required)):

The FERM domain is a protein-module that appears in more than 30 gene-products in the human genome. This domain enables protein-protein and protein-lipid interactions, which determine the localization, and consequently the activity and specificity of the individual FERM domain-containing protein. The FERM domain was thought initially to appear in cytoskeletal proteins with cytoplasmic/membranal localization, although some nuclear activities of such proteins have been reported over the past years.

Overall, this is an interesting article that presents the evolution of the FERM domain, which evolved already in bacteria and archaea, and is conserved up to primates. The article also presents a nice review of the structural features of the FERM domain, as well as the evolution of the nuclear localization signal that is responsible for its nuclear localization. Finally, the article covers in detail the properties and regulation of many FERM domain-containing proteins. Nonetheless, I have several minor concerns that should be addressed prior to publication. These are as follows:

1. Although the description of the individual FERM-domain containing proteins is of importance, the information on the role of the FERM domain in some of these proteins is not very focused. Since the review is about FERM domain, it is recommended to make it clearer.

ANSWER: Thank you for this comment. Wherever possible, we always tried to emphasize the role of the FERM domain, but we reworded and supplemented the parts in question in accordance with the suggestion.

2. *Some of the proteins seem to be exported from the nucleus by CRM1. How conserved is it, and how is the NLS/NES equilibrium determined?*

ANSWER: When collecting the data, we didn't focus enough on NES motives and transport regulation. Therefore, we once again thoroughly reviewed the literature from this point of view and were able to supplement the text and Figure 2. with additional information. E.g. we found two more, experimentally identified NES motifs and a proteomic screen identifying CRM1 binding partners (Kırlı et al, 2015). However, there is still very little data available to answer the questions raised by the reviewer (conservation and determination of equilibrium). CRM1 is the main nuclear exportin, with more than 1,000 potential binding partners (Kırlı et al, 2015), so it is not surprising that several FDCP proteins are transported from the nucleus by this exportin. Therefore, and given the small amount of data available, we would be wary of drawing conclusions about conservation of CRM1-mediated FDCP export. Relatively more data are now available on the NLS motifs (8 protein families out of the 18), and they show that the mechanisms of nuclear transport of FDCP proteins are certainly very diverse. This conclusion was discussed in the concluding remarks, but has been reformulated in response to the comment of the reviewer.

3. *Similarly, how conserved it's the NTS (as appears in Pyk2), and what are the relation of this sequence with the canonical NLS in this protein.*

ANSWER: We did not investigate NTS conservation because it is a very short motif that can occur relatively often in proteins. In addition, the motif is likely to play an accessory role and increase the rate of transport only when it is located proximal to the NLS motif EKRRK(E/R)(K/L/R/S/T), which is recognized by importin 7 (Panagiotopoulos et al, 2021). Interestingly, the sequence of Pyk2 does not contain this binding site for importin 7 or another canonical NLS near the NTS, so the exact mechanism of nuclear import of PYK2 has yet to be explored. We included this into the text.

4. *Is there an information on the properties of FERM-deleted proteins?*

ANSWER: This is an interesting question, and we assume that the reviewer is asking from the point of view of nuclear localization and function. In the absence of the FERM domain, the protein size in a significant proportion of FDCPs falls below the nuclear diffusion limit (40-60 kDa), so in these cases it is not possible to draw conclusions from localization. For larger proteins (FARPs, phosphatases, FRPDs, kinases, Talin, myosins, and KRIT), we reviewed the literature from this perspective, and the only data we found is that in the case of PTN3, the mutation of the three functional NLSs in the protein inhibits nuclear localization only if the FERM domain is removed at the same time, suggesting that the FERM domain retains the protein in the cytoplasm. This has been included into the text.

5. *The abstract is somewhat confusing. I had to read it several times to understand it. It is recommended to make it clearer.*

ANSWER: This has been done.

6. *The terms kinases and phosphates should be replaced to protein kinases and protein phosphatases as other type of kinases and phosphatases also exist but are probably not dependent on FERM domains.*

ANSWER: This has been corrected.

Reviewer #3 (Comments to the Authors (Required)):

In the present manuscript Borkuti et al. aim to review the nuclear localization and functions of FERM domain-containing proteins (FDCPs).

The authors claim that they had provided a comprehensive overview on the "exciting new dimension of the activity of FDCPs, their nuclear transport and function, with special regards to the role of the FERM domain in these processes". In fact, the review lists evolutionary development and conservation of FDCPs, with some hints on the mechanism of nuclear import/export domains, NLS NES etc and their localization in the context of the FERM domain. Insight into nuclear function of most proteins largely remains illusive, or is not described in much detail to really get a comprehensive overview about nuclear function. In most cases, putative interaction partners are listed, not even explained. Hence, the review is not comprehensive without inquiring additional literature. Many of the papers cited are 25-30 years old.

ANSWER: With this review, we would like to draw attention to the fact that in many cases the nuclear localizations of FDCPs were observed 2-3 decades ago (which is why we cite 25-30 years old articles), and that despite this, little or nothing is known about their exact nuclear functions today, much of the data on this is speculative. Therefore, it is not possible to create the detailed, comprehensive picture that the reviewer missed. However, the relatively few data available today indicate that these proteins are important players in nuclear function and that it could therefore obviously be an interesting new area of research in the future.

The reviewer's criticism highlighted that this basic message was not clear enough, so we reworded the title of the article (it was misleading) and rewrote both the abstract and the discussion. In addition, as suggested, we wrote about nuclear function for all proteins, even if nothing is known yet.

ERM

–*Detection in the nucleus*

-NLS motives present

-Novel ligands have been validated, among them nuclear proteins

-Function?...not described

ANSWER: Drosophila Moesin participates in mRNA export and regulates gene expression, but as far as we know, no other function has been described so far.

Merlin/NF2

-Contains NLS, NES and CRS sequences

-Undergoes nucleo-cytoplasmic shuttling in a cell cycle-dependent manner

-Inhibition of E3 ubiquitin ligase CRLDCAF1: function: restraining activity of Hippo signaling

-Interaction of Merlin with Lin28B (nucleolar RNA-binding protein: function?)

ANSWER: The following sentence has been added to the text: "The sequestration of Lin28B enables the maturation of pri-let-7 miRNAs, thereby Merlin inhibits cell growth."

-Binding of Merlin/NF2 to PICT-1: function?

ANSWER: The text has been supplemented with the following: "... and this interaction modulates the inhibitory effect of PICT-1 on the cell cycle and proliferation."

Protein 4.1

-Localizes diffusely in the nucleoplasm

-Nuclear binding partners point to a role in mRNA splicing and maintaining nuclear architecture - which and how?

ANSWER: The binding partners for maintaining nuclear architecture are Emerin, LaminA, NuMA, and actin. The binding partners of mRNA splicing are SC35 and U2AF35. To our knowledge, how they work together is still completely unknown.

-4.1R identified by mass spectrometry in nuclear envelope, associated with emerin and laminA...

ANSWER: The following sentence has been added to the text: "These results and the finding that depletion of 4.1R affects nuclear structures together suggest that the protein acts as a linker or adaptor at nodes vital for interconnections between nucleoplasmic substructures, nuclear envelope components and the nucleo-cytoskeletal interface (Meyer et al, 2011)."

-4.1N associates with NuMA preventing its role in mitosis - which?

ANSWER: The text has been supplemented with the following parts: “prevents its role in nuclear assembly at the end of mitosis” and “The interaction is necessary for the nuclear localization of P4.1 (Mattagajasingh et al, 2009) and nuclear assembly (Krauss et al, 2002), but the exact molecular mechanisms behind these activities of P4.1 are presently unknown.”

–Co-localization of protein 4.1 and actin in nuclei detected...

ANSWER: The text has been supplemented with the following sentence: “which also points to a structural function, although considering the manifold and not primarily structural functions of nuclear actin (Kloc et al, 2021), this observation suggests that the role of P4.1 in the nucleus might not be limited to structural tasks.”

–In rats binds to PIKE to prevent nuclear PI3K activity

–Localization in mRNA splicing factories, direct interaction with SC35 and U2AF35 (all citations about 25-30 years old!)

ANSWER: Please see the answer above.

FRMD

–Nuclear localization of FRMD6 in diverse cancers

–Co-localization with c-Met in glioblastoma - function?

ANSWER: The function and thus the significance of co-localization is not yet known. According to the authors' speculation FRMD6 may regulate c-Met functions such as calcium signaling in the nucleus.

FARP

–N-terminally truncated FARP becomes nuclear....

ANSWER: Human FARP2 was identified in a proteomics screen as a cargo for Exportin-1, but unfortunately, this is all the information that can be found in the literature about the nuclear localization of FARPs. This is mentioned now in the text.

Phosphatases

–PTN4, PTN13, PTN14 detected in nucleus

–PTN13 putatively inhibits STAT phosphorylation

–YAP is direct target of PTN14 - function?

ANSWER: The following sentence has been added to this paragraph: “These results indicate that PTPN14 in the nucleus suppresses the transcriptional coactivator activity of YAP and assists in the removal of inactive nuclear YAP, thereby inhibiting YAP-dependent cell migration.”

FARP

–Nothing really known

JAK

–Nuclear localization of JAK1, JAK2 and TYK2

–JAK2 co-localizes with chromosomes

–Direct interaction of JAK1 and JAK2 with STAT1, STAT3, and IFNGR1 in the nucleus

–Phosphorylation of STAT1 by JAK1 inhibits NES of STAT1

–JAK2 promotes gene expression by phosphorylation of H3

–Potential role of JAKs in maintenance of chromosome X inactivation...

FAK

–Acts as a scaffold to stabilize complexes of p53/MDM2 to promote p53 degradation, or GATA4/CHIPS for the same purpose

–Can directly regulate transcription: forms complexes with MBD2 that recruits the NURD complex to methylated CpG promotor sites, thereby promoting HDAC1 dissociation and inhibition of myogenin transcription

–Direct binding to: IL-33 and its receptor, ST2, TAF9, EZH2, RunX1 - what are the functions of these proteins and their interaction with FAK?

ANSWER: The text has been supplemented with the requested information and the entire paragraph has been rewritten.

–Detected in the nucleolus - protects Nucleostemin from proteasomal degradation - consequence?

ANSWER: The sentence has been rewritten. It reads now: “Active FAK was detected also in the nucleolus, where it protects Nucleostemin, a nucleolar GTPase that safeguards mitotic stem/progenitor cells from DNA damage in the S-phase, from proteasomal degradation (Tancioni et al, 2015), thereby promoting its function in ribosomal biogenesis and proliferation.”

Talin

–Interacts with chromatin - function unknown

Kindlin

–Kindlin2 in the nucleus of different mammalian cells

–Specifically interacts with the active form of β -catenin - function?

ANSWER: The following sentence has been added to the text: “The interaction selectively strengthens the occupancy of β -catenin on the Wnt target gene Axin2, and thereby promotes its expression.”

Myosins

–MyosinX present in nuclei of Xenopus

KRIT

–Nuclear import through ICAP1 α to prevent its proteasomal degradation in the cytosol

–Function?

ANSWER: The following sentence has been added to this part: “This result demonstrates the dual role of KRIT1 as a cytoplasmic and a nuclear protein, but its exact function in the nucleus remains at present completely unknown.”

January 19, 2024

RE: Life Science Alliance Manuscript #LSA-2023-02489R

Dr. Peter Vilmos
HUN-REN Biological Research Centre Szeged
Institute of Genetics
62. Temesvari krt.
Szeged 6726
Hungary

Dear Dr. Vilmos,

Thank you for submitting your revised manuscript entitled "FERM domain-containing proteins are active components of the cell nucleus". We would be happy to publish your paper in Life Science Alliance pending final revisions necessary to meet our formatting guidelines.

- please be sure that the authorship listing and order is correct
- please add the Twitter handle of your host institute/organization as well as your own or/and one of the authors in our system
- please upload a clean version of your manuscript file without tracking changes
- please add callouts for Figure 1A and B to your main manuscript text

A. FINAL FILES:

B. MANUSCRIPT ORGANIZATION AND FORMATTING:

Sincerely,

January 22, 2024

RE: Life Science Alliance Manuscript #LSA-2023-02489RR

Dr. Peter Vilmos
HUN-REN Biological Research Centre Szeged
Institute of Genetics
62. Temesvari krt.
Szeged 6726
Hungary

Dear Dr. Vilmos,

Thank you for submitting your Review entitled "FERM domain-containing proteins are active components of the cell nucleus". It is a pleasure to let you know that your manuscript is now accepted for publication in Life Science Alliance. Congratulations on this interesting work.

DISTRIBUTION OF MATERIALS:

Again, congratulations on a very nice paper. I hope you found the review process to be constructive and are pleased with how the manuscript was handled editorially. We look forward to future exciting submissions from your lab.

Sincerely,
